# Assessment of COVID-19 vaccine acceptance and its associated factors in Debre Berhan City, Ethiopia, 2022

**Eyuel Wubshet[1], Abinet Dagnaw[1], Helen Gebrelibanos[2], Mitiku Tefera[2] ***

**1** Department of Nursing, Debre Berhan Health Science College, Debre Berhan, Ethiopia, **2** Department of Public Health, School of Public Health Asrat Woldeyes Health Science Campus, Debre Berhan University, Debre Berhan, Ethiopia

* mitikutefera2632@gmail.com

## Abstract

### Background

The COVID-19 pandemic has spread over the world. The ability to achieve sufficient immunization coverage to end the global pandemic depends on the acceptance of the COVID-19 vaccine, but it has faced a major challenge around the world. In low-income and developing countries, 22.7% of the population has received at least one dose of the Covid-19 vaccine, which means that a large percentage of the population are unvaccinated, even though they have access to the Covid-19 vaccine so many countries do not accept the vaccine. The aim of this study was to assess COVID-19 vaccine acceptance and its associated factors in Debre Berhan City, Ethiopia, 2022.

### Methods

A mixed-methods approach comprising both qualitative interviews and a quantitative survey was used among participants in Debre Berhan City. A multi-stage sampling technique was used to recruit the study participants. An in-depth interview was used for the qualitative data. Data was collected by a face-to-face interview questionnaire from June 08 to July 08, 2022. The collected data was entered using Epi Data version 4.6 and analyzed using SPSS version 25. Variables with a p-value less than 0.25 at Bivariable logistic regression analysis were entered into multivariable logistic regression analysis. Logistics regression was employed, and a p-value <0.05 was considered statistically significant.

### Result

A total of 765 participants were included in the study, with a response rate of 97.08%. More than half (52.9%) of the respondents had the willingness to accept the COVID-19 vaccine. Participants who had Contact with COVID-19 patient (AOR = 3.98; 95% CI: (1.30–12.14), having good knowledge of COVID-19 vaccine (AOR = 4.63; 95% CI: (1.84–11.63), and positive attitude toward the COVID-19 vaccine (AOR = 3.41; 95% CI: (1.34–8.69) were statistically significantly associated variables with COVID-19 vaccine acceptance.

**Data Availability Statement:** All relevant data are within the paper and its Supporting Information files and additionally we will submit immediately when you ask

**Funding:** The author(s) received no specific funding for this work.

**Competing interests:** The authors have declared that no competing interests exist.

## Conclusion and recommendation

The present study revealed that the acceptance COVID-19 vaccine was 52.9, and a significant proportion of participants were hesitant to receive the vaccine and refused to get vaccinated. Significantly associated Variables for COVID-19 vaccine acceptance were Contact with COVID-19 patient, having good knowledge of the COVID-19 vaccine, and having a positive attitude towards the COVID-19 vaccine. Various stakeholders to apprise the public about the cause of the disease and the scientific development of vaccine in order to enhance acceptance of the vaccine.

## Introduction

The severe acute respiratory syndrome coronavirus virus 2 (SARS-CoV-2) causes Coronavirus Disease (COVID-19), an infectious disease. Most people infected with the virus experience mild to moderate respiratory illness and become seriously ill and require medical treatment. Older adults and people with comorbidities are more likely to develop serious illnesses. Anyone, at any age, can become seriously ill or die from COVID-19 virus [1, 2]. That being, many people were reluctant to get vaccinated due to false information about the COVID-19 pandemic on social media, including claims that the virus is connected to 5G cell phone networks, that vaccinations cause early death, and that the pandemic is a terrorist weapon [3].

The COVID-19 vaccine is effective in reducing the risk of catching and spreading the COVID-19 virus, and the vaccine can also help to avoid serious illness [4]. The ability to achieve sufficient immunization coverage to end the global pandemic depends on widespread acceptance of COVID-19 vaccines. Understanding the factors that influence COVID-19 vaccine acceptability is important since a delay in immunization might lead to the spread of novel variations that can overcome vaccine-induced immunity [5, 6].

The World Health Organization (WHO) has identified COVID-19 vaccination hesitation and refusal as one of the top 10 health dangers for 2019. This hesitancy and refusal are important concerns globally [7]. Every year, vaccines prevent 3.5–5 million deaths from infectious diseases. The development of a safe and effective COVID-19 vaccine is a major step forward towards ending the pandemic [8, 9]. To put an end to the pandemic, a huge portion of the population must be immune to the virus [10]. The globe has faced a major barrier in terms of willingness to accept a COVID-19 vaccine [11]. Lack of knowledge and confidence in vaccination is also currently recognized as the greatest threat to the success of vaccination programs [12]. According to the Africa Centers for Disease Control and Prevention (CDC), the COVID-19 vaccine utilization in Ethiopia is 22.09% [13]. The effectiveness of the COVID-19 vaccination effort is determined by the percentage of the population willing to be vaccinated, and recent estimates predict that vaccination of up to 70% of the population may be required to stop the current pandemic [14, 15].

A cross-sectional study conducted in nine countries with middle and low incomes found that higher income, younger age, and female gender had statistically significant effects on vaccine acceptance [16].Some myths and conspiracy theories about vaccine and COVID-19 affect the acceptance of COVID-19. High COVID-19 vaccination awareness and individuals with a favorable attitude towards the COVID-19 vaccine were statistically significantly associated with the acceptance of the COVID-19 vaccine in Ethiopia [11, 17]. Therefore, this study explored the extent of COVID-19 vaccine acceptance among residents over the age of eighteen

in Debre-Berhan city, Amhara region, Ethiopia, and the factors that enabled them not to accept the vaccine through qualitative and quantitative research methods.

According to systematic studies, the pooled proportion of COVID-19 vaccine acceptance in Ethiopia is 56.02% [18].

## Methods

### Study design, setting, and period

Community-based cross-sectional mixed-method approach comprising both qualitative interviews and a quantitative study was conducted from June 08 to July 08 2022. The study was conducted in Debre Berhan City, Debre Berhan was founded by the great king and philosopher "Emperor" Zerayakob. In the 15th century, Debre Berhan was one of the oldest cities in the world and was the capital of Ethiopia. Debre Berhan Locate in the north shoa zone of Amhara region About 130km far from Addis Ababa with an elevation of 2840 Meters which makes it the highest city in Africa, far from Bahirdar (The Regional Capital city) about 518 km.

**Source population.**
✓ All adult population over the age of 18 years who are living in Debre Berhan City
**Study population.**
✓ All adult population over the age of 18 years in the selected kebele
**Eligibility criteria.**
✓ *Inclusion criteria.*
✓ Who lived at least 6 months in the City
✓ aged >18 years was included in this study
*Exclusion criteria.*
✓ People who were seriously ill during the period of data collection
✓ Who are unable to communicate

**Sample size determination.** The required sample size (n) for the quantitative part of the study was determined using a single population proportion formula considering the following assumptions. A similar study about the COVID-19 vaccine acceptance in the southern region of Ethiopia found P = 62.6 [11].

$$\text{Confidence level}(Z) = 95\%; 5\% \text{margin of error(d)}; n = \frac{(1.96)^2 \times (0.63) \times (1 - 0.6)}{(0.05)^2}$$

$$\mathbf{n = 358}$$

Finally, because of the 2 design effects, the minimum sample size for this study will be 716. By adding a 10% non-response rate, the total sample size becomes **788.**

The study participants were recruited using a multi-stage sampling technique. Using a simple random sampling technique, three kebele were chosen randomly by lottery method from nine kebele. Households were identified in the family folders of health extension staff (HEWs), and study participants were chosen using a systematic random sampling process. Hence, a systematic random sampling technique was done by determining the K value by dividing the source population by the calculated sample size, and then by accustoming this selected number to each K value(k = 7), recruited the study participants (show Fig 1).

For the qualitative part, 16 participants were included in the study based on information saturation.

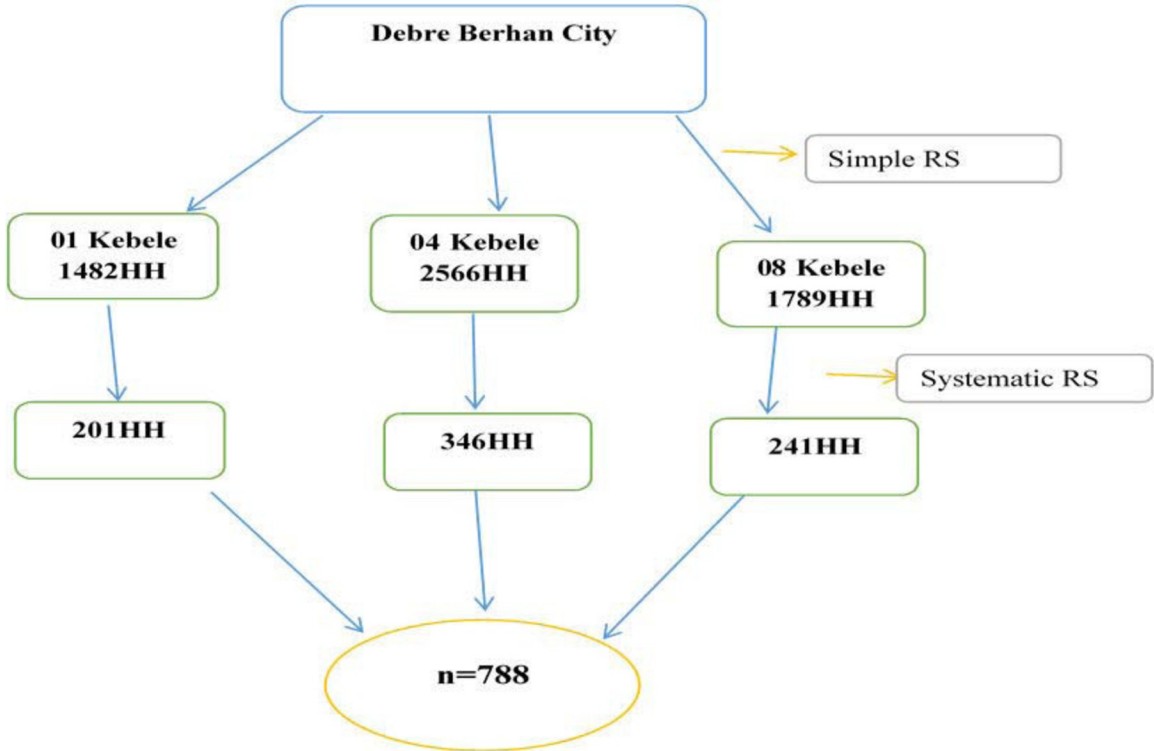

**Fig 1. Schematic presentation of sampling procedure on acceptance of covid-19 vaccine and its associated factors among in Debre Berhan City.**

### Study variables

**Dependent variable.**
✓ The acceptance of the COVID-19 vaccine
**Independent variables.**   Socio demographic data–
✓ Knowledge, and attitude related factor
✓ Health related factors–
✓ Vaccine related factors -

**Data collection procedure.**   The data were collected using a face-to-face interviewer-administered semi-structured questionnaire, which was developed from reviewed literature [11, 19–21]. The data were collected by six nurses who have a diploma and one supervisor who has a Master's degree in public health. The data collection tool had four subsections which are socio-demographic questionnaire, knowledge and attitude, health-related factors and the fourth was vaccine-related factors. For the qualitative portion of the study, 16 people who were specifically chosen by the investigator were subjected to in-depth interviews.

**Data quality assurance.**   The questionnaires were prepared in the English version and translated into an Amharic version, and again back to the English version to check for inconsistency in the meaning of words. Before data collection, one-day training was given by the principal investigator to the data collectors and supervisor about the objective of the study, the data collection tool, and the data collection process. To avoid data contamination, 5% of the sample size was pre-tested in Chahca town before the actual data collection period began. The data collection tool was checked for completeness, clarity, and appropriateness during the pre-test, by the supervisor and principal investigator.

**Data processing and analysis.** The filled questionnaire was coded and entered into Epi Data version 4.6 software. For analysis, the entered data was exported to the SPSS version 25 software. For the result part, descriptive statistics (frequency, proportion, and mean) are used to summarize data and evaluate the distribution of responses. The data reliability, normality, and multicollinearity were checked. The association between the independent variable and the outcome variables was determined using multivariable logistic regression. Model fitness was checked by the Hosmer–Lemeshow test. Crude odds ratios with a 95% confidence interval were calculated for all variables included in the Bivariable model. A multivariable analysis was performed for a variable that has a p-value less than 0.25 to determine the independent effects of each variable. A p-value of less than 0.05 in the final model was taken as a statistically significant factors. The adjusted odd ratio (AOR) from multivariable logistic regression was used to measure the strength of association between dependent and independent variables. Finally, the findings are summarized through the use of frequency distributions, texts, tables, and graphs.

The qualitative data analysis was done by transcription and translation of the interviews, then coded and analyzed by thematic analysis, and statements from participants were presented verbatim to illustrate the themes realized. The findings of the qualitative study were used to supplement the findings of the quantitative data.

**Operational definitions.** *Covid-19 vaccine acceptance.* "Have you been vaccinated, and will you get vaccinated if you get the COVID-19 vaccine?" Those who respond "Yes" to these question were considered vaccine accepted and those who respond "No" were considered vaccine hesitant [20].

*Knowledge and attitude towards Covid-19 vaccine.*

✓ Good knowledge: - refers to respondents with individual knowledge scores of mean and above [11].

✓ Poor knowledge: - refers to respondents with individual knowledge scores below the mean score [11].

✓ positive Attitude: Refers to those study participants who scored greater than or equal to the mean of attitude questions [11].

✓ Negative Attitude: Refers to those study participants who scored less than the mean of attitude questions [11]

## Ethical clearance

Ethical clearance was obtained from Asrat Woldeyes, Health Science Campus, on behalf of the institutional Review Board of Debre Berhan University (Protocol number, IRB-022). A supportive letter was also obtained from Debre Berhan City municipal office to all selected Kebele's and administrative offices. Each study participant was adequately informed about the purpose, method, and anticipated benefits and risks of the study by the data collector. Respondents have the right to respond or refuse the interview. Written consent was received from study participants. The information was kept confidential by not writing their names, and there was no incentive or compensation for study participants.

## Results

### Socio-demographic characteristics

A total of 765 participants were involved in this study, with a response rate of 97.08%. Over half (55%) of participants were females. The mean age of the participants was 37.8 (SD ±15.3) years, with a minimum age of 18 years and a maximum age of 81 years. In terms of marital status, 260 (34%) of the study participants were unmarried, followed by married 365 (47.5%). In terms of occupation, 116 (15.2%) were unemployed, followed by government employees and

**Table 1. Socio-demographic characteristics of the study participants of COVID-19 vaccine acceptance and its associated factors in Debre Berhan City, Ethiopia, 2022.**

| Variables | Category | Frequencies (n = 765) | Percentages (100%) |
|---|---|---|---|
| **Age** | 18–25 | 211 | 27.6 |
| | 26–35 | 191 | 25.0 |
| | 36–45 | 144 | 18.8 |
| | >46 | 219 | 28.6 |
| **Sex** | Male | 344 | 45 |
| | Female | 421 | 55 |
| **Marital Status** | Single | 260 | 34 |
| | Married | 363 | 47.5 |
| | Other*** | 142 | 18.5 |
| **Educational status** | Unable to read and write | 29 | 3.8 |
| | Primary school | 123 | 16.1 |
| | Secondary school | 211 | 27.6 |
| | College and above | 402 | 52.5 |
| **Occupational Status** | Civil Servant | 236 | 30.8 |
| | Unemployed | 116 | 15.2 |
| | Merchant | 174 | 22.7 |
| | Student | 115 | 15.1 |
| | Other** | 124 | 16.2 |

*** For widowed divorced and separated

** for daily laborer and NGO employee

private businesses (30.8% and 22.7%). More than one-fourth (27.6%) of study participants had a secondary education, 402 (52.5%) had a college or higher level of education (Table 1).

**Acceptance towards Covid-19 vaccine.** The acceptance of the COVID-19 vaccine was 405(52.9%) among respondents, and 360 (47.1%) were hesitant to receive the COVID-19 vaccine (Fig 2).

**Respondent's reason for hesitancy for Covid-19 vaccine.** Of the 446 (58.3) participants who were not taking the COVID-19 vaccine, from those 177 (39.7%) were afraid of adverse effects of the COVID-19 vaccine, and 162 (36.6%) preferred other ways of protection for the disease and inadequate information about the safety of the vaccines.

**Reasons for Covid-19 vaccine hesitancy.** In the qualitative in-depth interview findings, there were three factors list for COVID-19 vaccine hesitancy.

*Theme 1. Individual factors.* Some participants stated they had not had childhood vaccinations in their lifetime. Furthermore, a number of the study participants had a belief that they were young and in good health, which led them to negative reception of the vaccine.

*"The COVID-19 vaccination is ineffective. I am aware that people who receive the immunization still contract the COVID virus." (Participant 7)*

*"I will not take the COVID-19 vaccine because my body does not accept it. And I have not given any vaccinations, even Child Routine immunization to my four children and they are healthy "(Participant 10 and 12)*

*"Since I am young and in good health, I don't need to get immunized by COVID-19." (Participant 13)*

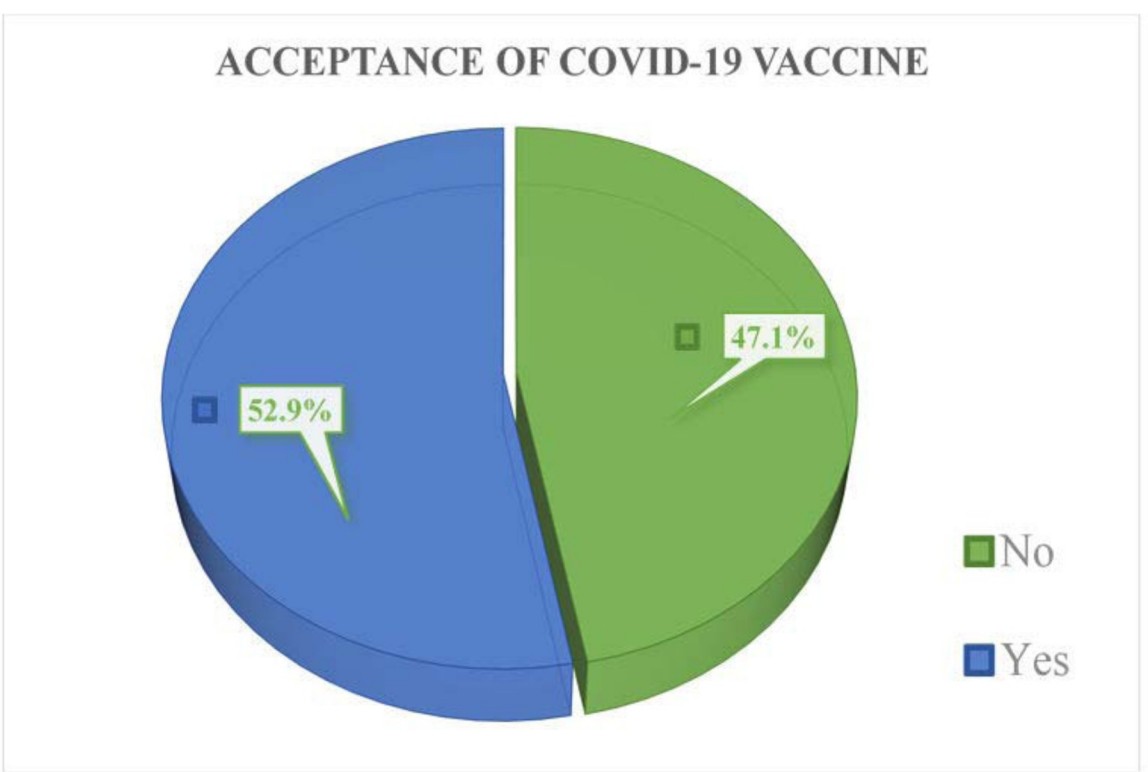

**Fig 2. Respondents acceptance to covid-19 vaccine in a study of Covid-19 vaccine acceptance and its associated factors in Debre Berhan City.**

*Theme 2. Concerns over possible side effects.* The fear of adverse reactions to vaccines was one of the key factors contributing to vaccine hesitation, which explained why some people were postponing being vaccinated:

*"I have heart and kidney problems. When I take medicine, my body swells up, and if I go, I go to Tsebel to pray. He will save me. I will never take the coronavirus vaccine." (Participant 6)*

*"I got sick after being given trachoma medicine and then I lost faith in any vaccine, so I don't want to take the corona vaccine." (Participant 3 and 5)*

*"I don't want to take the corona vaccine because I have heard that it causes blood clots and I am worried about vaccine side effects." (Participant 1 and 4)*

*Theme 3. Socio cultural factors.* Additionally, it was claimed that the vaccinations might be used as a weapon as the beast's mark, or "(666)," which would lead people to turn away from their faith. Others believed they were protected by God and did not require the vaccine.

*"My faith in God will protect me, so I don't need to get vaccinated." (Participant 8, 15, 11 and 14)*

*"Getting vaccinated by COVID-19 is taking the beast sign of 666." (Participant 16, 2 and 9)*

**Table 2. Respondent's health related factors towards COVID-19 of a study of COVID-19 vaccine acceptance and its associated factors in Debre Berhan City, Ethiopia, 2022.**

| Variables | Category | Frequency | Percent (%) |
|---|---|---|---|
| Do you have history of chronic illness? | Yes | 253 | 33.1 |
| | No | 512 | 66.9 |
| Hypertension | Yes | 93 | 63.2 |
| | No | 160 | 36.8 |
| Diabetes | Yes | 59 | 23.3 |
| | No | 194 | 76.7 |
| Heart disease | Yes | 71 | 28.1 |
| | No | 182 | 71.9 |
| Stroke | Yes | 16 | 6.3 |
| | No | 237 | 93.7 |
| HIV/AIDS | Yes | 36 | 14.2 |
| | No | 217 | 85.8 |
| Abdominal pain | Yes | 45 | 17.8 |
| | No | 208 | 82.2 |
| Have you been previously infected with COVID-19? | Yes | 119 | 15.6 |
| | No | 646 | 84.4 |
| Have you ever contacted someone with a COVID-19 patient? | Yes | 157 | 20.5 |
| | No | 608 | 79.5 |
| Have you had laboratory test for COVID-19 disease? | Yes | 232 | 30.3 |
| | No | 533 | 69.7 |
| What was the result of COVID-19 Test? | Negative | 108 | 46.2 |
| | Positive | 126 | 53.8 |

**Health related factors towards Covid-19 vaccine.** One third (33%) of the participants had chronic illness. predominantly, hypertension, diabetes, and heart disease. 119 (15.6%) of the participants had previously been infected with COVID-19. 157 (20.5%) of the participants had met with COVID-19 patient. 232 (30.3%) of the participants had laboratory tests for COVID-19 disease, from those 108 (46.2%) of participants had a positive result of the COVID-19 Test (Table 2).

**Knowledge towards Covid-19 vaccine.** Four hundred sixty-three, (60.5%) of the participants, knew about the COVID-19 vaccine. More than half (50.7%) of the respondents knew about the effectiveness of the COVID-19 vaccine. Almost three-quarters (73.8%) of the study participants, said an overdose of the COVID-19 vaccine would be dangerous to humans. Three-fourths (61.3) of the respondents explained that COVID-19 vaccination did not increase allergic reactions. Six hundred forty (80.3%) of the participants explained that COVID-19 vaccination would not increase the risk of other diseases. Half (50.3) of the participants had poor knowledge about the COVID-19 vaccine (Table 3).

**Attitude towards Covid-19 vaccine.** More than half (52.7%) of the participants agreed that the newly discovered COVID-19 vaccine was safe. Almost 415 (54.2%) of the participants agreed that the COVID-19 vaccine was essential for us. Almost 403 (52.6%) of the respondents agreed to encourage family/friends/relatives to get vaccinated. More than half of the respondents (53.2%) agreed that the COVID-19 vaccine can prevent COVID-19 infection. Around half of the respondents had a Negative attitude toward the COVID-19 vaccine (Table 4).

**Vaccine related factors towards Covid-19.** Half (50.6%) of the participants had received all the necessary vaccinations in their lifetime. The majority of the respondents 509 (66.6%) of the participants had trust in the health system regarding to the COVID-19 vaccine. The

**Table 3. Respondent's knowledge towards COVID-19 of a study COVID-19 vaccine acceptance and its associated factors in Debre Berhan City, Ethiopia, 2022.**

| Variables | Category | Frequency | Percent (%) |
|---|---|---|---|
| Do you know about the COVID-19 vaccine? | Yes | 463 | 60.5 |
| | No | 302 | 39.5 |
| Do you know about the effectiveness of the COVID-19 vaccine? | Yes | 388 | 50.7 |
| | No | 377 | 49.3 |
| Is it dangerous to use an overdose of COVID-19 vaccines? | Yes | 565 | 73.9 |
| | No | 200 | 26.1 |
| Does COVID-19 vaccination increase allergic reactions? | Yes | 296 | 38.7 |
| | No | 469 | 61.3 |
| Does taking COVID-19 increase your risk of other diseases? | Yes | 151 | 19.7 |
| | No | 614 | 80.3 |

majority of the respondents 548 (71.6%) had used more than two COVID-19 preventive measures. 357 (46.7%) of the participants had used one source of information regarding the COVID-19 vaccine (Table 5).

## Factors associated with Covid-19 vaccine acceptance

For analysis of the data, Bivariable and multivariable logistic regression were done by using binary logistic regression. The crude and adjusted odds ratios with a 95% confidence interval were calculated to determine the strength of association and statistical significance between acceptance of the COVID-19 vaccine and each independent variables.

In the Bivariable logistic regression analysis, the following variables were found to have a p-value less than 0.25. These variables were age, marital status, educational status, occupation, income, chronic disease, Contact with COVID-19 patient, laboratory test for COVID-19 disease, the result of the COVID-19 test, knowledge of COVID-19, attitude towards COVID-19 vaccine, who received all the necessary vaccinations in their lifetime, and who had trust in the health system regarding the vaccine. All these variables were entered into the multivariable analysis.

**Table 4. Respondents attitude towards COVID-19 vaccine of a study on COVID-19 vaccine acceptance and its associated factors in Debre Berhan City, Ethiopia, 2022.**

| Variable | Category | Frequency | Percent (%) |
|---|---|---|---|
| Is newly discovered COVID-19 vaccine safe? | Agree | 403 | 52.7 |
| | Disagree | 362 | 47.3 |
| COVID-19 vaccine is essential for us? | Agree | 415 | 54.2 |
| | Disagree | 350 | 45.8 |
| COVID vaccine developed in Europe and America are safer than those made in other world countries? | Agree | 386 | 50.4 |
| | Disagree | 379 | 49.6 |
| May you encourage your family/friends/relatives to get vaccinated? | Agree | 403 | 52.6 |
| | Disagree | 362 | 47.4 |
| COVID-19 vaccine can prevent covid-19 infection? | Agree | 407 | 53.2 |
| | Disagree | 358 | 46.8 |
| It is not possible to reduce the incidence of COVID-19 without vaccination? | Agree | 382 | 49.9 |
| | Disagree | 383 | 50.1 |
| The COVID-19 vaccine should be distributed fairly to all of us? | Agree | 437 | 57.1 |
| | Disagree | 328 | 42.9 |

**Table 5. Respondents vaccine related factors towards COVID-19 vaccine of a study on COVID-19 vaccine acceptance and its associated factors in Debre Berhan City, Ethiopia, 2022.**

| Variable | Category | Frequency | Percent (%) |
|---|---|---|---|
| Have you received all the necessary vaccination in your lifetime? | Yes | 387 | 50.6 |
|  | No | 378 | 49.4 |
| Do you have Trust in the health system regarding to the vaccine? | Yes | 509 | 66.5 |
|  | No | 256 | 33.5 |

Among them, three variables were significantly associated with the outcome variables, i.e., acceptance of the COVID-19 vaccine at a p-value 0.05. In the multivariate analysis, Contact with COVID-19 patient, having good knowledge of COVID-19, and positive attitude towards COVID-19 were found to be significantly associated with acceptance of the COVID-19 vaccine.

From health-related factors, Contact with COVID-19 patient was found to be significantly associated with acceptance of the COVID-19 vaccine. The finding showed that Contact with COVID-19 patient was nearly four times more likely to accept COVID-19 vaccine compared with those who have never Contacted with COVID-19 patient (AOR = 3.98; 95% CI: 1.30–12.14).

Furthermore participant's good knowledge of COVID-19 vaccine was found to be significantly associated with acceptance of the vaccine. The finding showed that Participants with the good knowledge of the COVID-19 vaccine were five times willing to accept COVID-19 vaccine compared with those who had poor knowledge of the COVID-19 vaccine. (AOR = 4.63; 95% CI: 1.84–11.63).

Finally, participants' Positive attitude towards the COVID-19 vaccine was found to be significantly associated with acceptance of the vaccine. The finding showed that the odds of accepting the COVID-19 vaccine were three times greater for those who had a positive attitude towards the vaccine than for those who had negative attitudes towards the vaccine. (AOR = 3.41; 95% CI: 1.34–8.69) (Table 6).

## Discussion

This study aims to assess COVID-19 vaccine acceptance and the associated factors in Debre Berhan City. Therefore, the result of this study found that 52.9% (95% CI; 49.4–56.4%) of participants were willing to accept the COVID-19 vaccine This finding is consistent with the findings of pooled results of the systematic review in Ethiopia, systematic review and meta-analysis studies conducted in Africa and the study conducted in Zimbabwe, which reported that 56%, 48.93% and 49.9% of participants were willing to accept the COVID-19 vaccination, respectively [18, 22, 23].

On the other hand the result of this study, the acceptance of COVID-19 vaccine was lower than study done in Dessie (59.4%) Gurage Zone (62.6%), wolkite (70.7%) and Debre berhan (69.3%) [11, 17, 24, 25]. The possible reason for the inconsistency might be that the source population for the Dessie and Wolkite studies were chronic patients and pregnant women attending antenatal care respectively, whereas the Debre Berhan's study was done among university students, while this study interviewed the general population in the community.

The result of this study was higher than the study done in; Addis Ababa (39.7%), Debre Tabor (42.3%), and the Wolaita Zone (45.5%) (25, 32, 56). This differences might be due to the differences in the sample size and study period.

The finding of this study is lower than the studies conducted in the United States (67%), and the United Kingdom showed (71.7%), China (91.3%), Japan (65.7%), Pakistan (70.2%), Saudi Arabia (64.7%), and Lebanon (63.4%) % of participants were willing to accept the

**Table 6. Bivariable and multivariable logistic regression analysis of associated factors in a study on COVID-19 vaccine acceptance and associated factor in Debre Berhan City, Ethiopia, 2022.**

| Variable | Category | Willingness to Accept COVID-19 Vaccine | | COR (95% CI) | AOR (95% CI) | p-value |
|---|---|---|---|---|---|---|
| | | Yes | No | | | |
| Age | 18–25 | 74(18.3%) | 137(38.1%) | 1 | 1 | |
| | 26–35 | 96(23.7%) | 95(26.4%) | **1.87(1.25–2.79)** | 1.23(0.32–4.65) | 0.75 |
| | 36–45 | 75(18.5%) | 69(19.2%) | **2.01(1.30–3.10)** | 0.81(0.81–3.51) | 0.78 |
| | >46 | 160(39.5) | 59(16.4%) | **5.02(3.32–7.57)** | 0.93(0.93–0.19) | 0.93 |
| Marital status | Single | 90(22.2 | 170(47.2%) | 1 | 1 | |
| | Married | 206(50.9 | 157(43.6%) | **2.47(1.78–3.44)** | 1.86(0.61–5.70) | 0.27 |
| | Others* | 109(26.9 | 33(9.2%.) | **6.23(3.91–9.94)** | 2.04(0.38–10.92) | 0.40 |
| Education | Unable read and write | 16(4.0%) | 13(3.6%) | 1.42(0.67–3.04) | 3.94(0.38–40.1) | 0.24 |
| | Primary school | 59(14.6%) | 64(17.8%) | 1.07(0.71–1.60) | 0.75(0.20–2.80) | 0.67 |
| | Secondary school | 144(35.6%) | 67(18.6%) | **2.49(1.75–3.54)** | 1.19(0.43–3.29) | 0.73 |
| | College and above | 186(45.9% | 216(60%) | 1 | 1 | |
| Occupation | Unemployed | 52(12.8%) | 64(17.8%) | 1 | 1 | |
| | Civil Servant | 140(34.6%) | 96(26.7%) | **1.79(1.14–2.81)** | 1.62(0.35–7.50) | 0.53 |
| | Merchant | 97(24.0%) | 77(21.4%) | 1.55(0.96–2.48) | 1.24(0.25–6.14) | 0.78 |
| | Student | 34(8.4% | 81(220.5%) | **0.51(0.30–0.88)** | 1.61(0.26–9.87) | 0.60 |
| | Other** | 82(20.2%) | 42(11.7%) | **2.40(1.42–4.04)** | 2.50(0.43–14.5) | 0.30 |
| Income | <1999 | 129(31.9%) | 184(51.1%) | 1 | 1 | |
| | 2000–3999 | 119(29.4 | 89(24.7) | **1.90(1.33–2.72)** | 0.65(0.17–2.46) | 0.53 |
| | >4000 | 157(38.8) | 87(24.2%) | **2.57(1.82–3.63)** | 0.93(0.24–3.61) | 0.92 |
| Chronic Disease | No | 212(52.3%) | 300(83.3) | 1 | | |
| | Yes | 193(41.7%) | 60(16.7%) | **4.55(3.24–6.38)** | 1.75(0.64–4.82) | 0.27 |
| Contact someone with COVID-19 patient | No | 275(67.9%) | 333(92.5%) | 1 | | |
| | Yes | 130(32.1%) | 27(7.5) | **5.83(3.73–9.09)** | **3.98(1.30–12.14)** | **0.01** |
| Laboratory test for COVID-19 disease | No | 222(54.8%) | 311(88.4%) | 1 | | |
| | Yes | 183(45.2%) | 49(13.6%) | **5.23(3.65–7.49)** | 1.74(0.06–48.6) | 0.74 |
| Result of COVID-19 Test | Negative | 96(52.2%) | 12(24.0%) | 1 | | |
| | Positive | 88(47.8%) | 38(76.0%) | **3.45(1.69–7.03)** | 0.94(0.32–2.73) | 0.91 |
| knowledge of COVID-19 | Poor | 115(28.4%) | 270(75.0%) | 1 | 1 | |
| | Good | 290(71.6%) | 90(25.0%) | **7.56-(5.48–10.43)** | **4.63(1.84–11.63)** | **0.001** |
| Attitude of COVID-19 vaccine | Negative | 76(18.8%) | 243(67.5%) | 1 | 1 | |
| | Positive | 329(81.2%) | 117(32.5%) | **8.99(6.44–12.54)** | **3.41(1.34–8.69)** | **0.01** |
| Received the necessary vaccination in lifetime | No | 141(34.8%) | 237(65.8%) | 1 | 1 | |
| | Yes | 264(65.2%) | 123(34.2%) | **3.60(2.67–4.86)** | 1.22(0.45–3.26) | 0.68 |
| Trust in the health system | No | 52(12.8%) | 204(56.7%) | 1 | 1 | |
| | Yes | 353(87.2%) | 156(43.3%) | **8.87(6.20–12.69)** | 2.27(0.76–6.81) | 0.14 |

1 indicates reference variables & Bold is for variables with p<0.05

COVID-19 vaccine [20, 26–31] respectively. These discrepancies might be related to differences in the method they used and the study settings and countries' socio-demographic; the availability and accessibility of the health service infrastructure; socio-economic and knowledge differences; and also the high death rate in those countries.

The Americas and China were among the most severely affected countries with the pandemic, with high rates of death per capita in the region. COVID-19 vaccine acceptance rates were generally high, with countries having >70% COVID-19 vaccine acceptance rates, which

may have reduced levels of hesitancy [32]. There is high coverage of the COVID-19 vaccine in developed countries, which is influenced by a positive attitude toward vaccination compared to developing countries.

In this study, contact someone with COVID-19 patient was found to be significantly associated with acceptance of the COVID-19 vaccine. This finding is in line with the study conducted in the Wolaita zone where study participants had contact with family and friends with COVID-19 [19]. The COVID-19 virus and its effects on health may have come to the attention of those who have COVID-19 virus. Those who had been diagnosed with the COVID-19 virus led them to seek protection from the COVID-19 vaccine.

The other factor that was found to be significantly associated with acceptance of the COVID-19 vaccine was having good knowledge of the COVID-19 vaccine. This might be because people who have good knowledge of the COVID-19 vaccine may also know that method of disease prevention and the possible complications of the diseases so that they will be able to understand the benefits of the COVID-19 vaccine. This finding is in line with the study conducted in Lebanon [31] Gurage zone, Ethiopia [11] and Dessie, Ethiopia [17]. There is evidence that the public's acceptance of vaccines is improved by the dissemination of information and education [33].

Finally, having a positive attitude towards the COVID-19 vaccine was found to be significantly associated with acceptance of the vaccine. This finding is in line with the study conducted in Dessie [17]. The possible explanation could be continued awareness creation by the government and non-governmental organizations about the importance of the COVID-19 vaccine has delivered.

The participant's acceptance and hesitancy to receive the vaccine varied according to the qualitative findings. The study shows misunderstandings about vaccines negatively impact vaccine willingness. The participants also mentioned that they preferred other COVID-19 preventive measures and believed that the illness was a punishment from God for our sins and that the only way to end it was by prayer and repentance. However, some of them connected it to the so-called satanic (666) religious act. These findings are consistent with a study conducted in Northeast Ethiopia [34].

## Conclusion

In this study the acceptance of the COVID-19 vaccine was (52.9%), and 47.1% of participants were found hesitant towards receiving the COVID-19 vaccine was available. COVID-19 vaccine acceptance is significantly associated with Contact someone with COVID-19 patient, having good knowledge of the COVID-19 vaccine, and having a positive attitude towards the COVID-19 vaccine.

The qualitative finding revealed that the main reasons for vaccine hesitation were negative attitudes about the vaccine, as well as individual and socio-cultural factors.

### Recommendations

Based on the findings, the following recommendations for different stakeholders are suggested:

The Ministry of Health needs to better to strengthen collaboration with various stakeholders to update the public through the media about the cause of the disease and the scientific development of the COVID-19 vaccine in order to enhance public acceptance of the vaccine. Policymakers and program managers better give emphasis and play an important role in the implementation of a comprehensive program that will improve the community's attitude and knowledge towards COVID-19 vaccine acceptance.

For future researchers: better to explore unanswered issue towards factors affecting vaccine acceptance. Religious leaders have to teach that the COVID-19 vaccination does not conflict with faith.

## Limitation

The study relied on verbal reports by the participants, which has the potential to result in recall bias and over-reporting or under-reporting of socially acceptable and unacceptable behaviors, respectively. The researcher tried to reduce this limitation by explaining to the respondents that their genuine responses are more valuable.

## Supporting information

**S1 File. *English* version questioners.**
(DOCX)

**S2 File.**
(ZIP)

## Acknowledgments

We would like to express our gratitude to Debre Berhan University, study participants, and data collectors for their sincere aid in bringing the necessary information.

## Author Contributions

**Conceptualization:** Abinet Dagnaw, Helen Gebrelibanos, Mitiku Tefera.

**Software:** Eyuel Wubshet, Abinet Dagnaw.

**Supervision:** Eyuel Wubshet, Abinet Dagnaw, Helen Gebrelibanos, Mitiku Tefera.

**Validation:** Eyuel Wubshet, Abinet Dagnaw, Helen Gebrelibanos, Mitiku Tefera.

**Visualization:** Eyuel Wubshet, Abinet Dagnaw, Helen Gebrelibanos, Mitiku Tefera.

**Writing – original draft:** Eyuel Wubshet, Abinet Dagnaw, Helen Gebrelibanos, Mitiku Tefera.

**Writing – review & editing:** Eyuel Wubshet, Abinet Dagnaw, Helen Gebrelibanos, Mitiku Tefera.

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
