## [Decision Letter · Decision Letter 0]

27 Mar 2023

PONE-D-23-01580Assessment of COVID-19 Vaccine Acceptance and its associated factors in Debre Berhan City, Ethiopia, 2022PLOS ONE

Dear Dr. Afessa,

Thank you for submitting your manuscript to PLOS ONE. After careful consideration, we feel that it has merit but does not fully meet PLOS ONE’s publication criteria as it currently stands. Therefore, we invite you to submit a revised version of the manuscript that addresses the points raised during the review process.

We look forward to receiving your revised manuscript.

Kind regards,

Ayechew Ademas Tesema, MSc

Academic Editor

PLOS ONE

4. Please amend the manuscript submission data (via Edit Submission) to include author Mitiku Tefera.

5. Please amend your authorship list in your manuscript file to include author Niguse Afessa.

6. Please include a caption for figures 1 and 2.

Additional Editor Comments:

Dear author, the manuscripit will have valuable contribution, so please edit the whole document gramatical errors. Edit and replace "others" in tables with specific variables with parenthesis and comments from the reviewers.

Reviewers' comments:

Reviewer's Responses to Questions

**Comments to the Author**

1. Is the manuscript technically sound, and do the data support the conclusions?

Reviewer #1: Yes

Reviewer #2: Partly

2. Has the statistical analysis been performed appropriately and rigorously? 

Reviewer #1: Yes

Reviewer #2: No

3. Have the authors made all data underlying the findings in their manuscript fully available?

Reviewer #1: Yes

Reviewer #2: Yes

4. Is the manuscript presented in an intelligible fashion and written in standard English?

Reviewer #1: No

Reviewer #2: No

5. Review Comments to the Author

Reviewer #1: Title; Assessment of COVID-19 Vaccine Acceptance and its associated factors in Debre Berhan City, Ethiopia, 202

The title; the title of this study is timely and good.

Abstract; the authors should summarize the challenge of COVID-19 and mention why they needed to conduct this study,

The objective of the study should be mentioned.

Better to summarize the methods in short and precise way. Better to use general terms to conclude the main findings of the research.

INTRODUCTION; the introduction parts lacks basic information regarding COVID 19 vaccine acceptance and its challenge, literature review, the burdens of the problems and the expected outcome should also be well stated,…

What is the need of putting specific objectives?

The authors should explain about the study area…..

In the inclusion criteria, why the authors need to include individuals who lived > 6months?

The sample size determination procedure is not clear……how the study subjects were selected?

Figure 1: Schematic presentation of sampling procedure, where is figure 1?

In the result part better to focus on the main findings rather than narrating the results.

The discussion part should focus on the findings and the factors associated with COVID 19 vaccine acceptances.

In general the manuscript needs grammar, coherence and spelling corrections.

Reviewer #2: Grammatical, punctuation, spelling, and sentence fragmentation have to be revised in the whole document.

Based on the objective and design of the study, your paper is a mixed-methods study. But the qualitative part of the study is not well addressed. - in the abstract section, the method section, the result and discussion sections, or even in the conclusion and recommendation sections.

What type of analysis do you conduct to integrate the qualitative results into the qualitative results?

Is it a balanced approach to QAN, both qualitatively and quantitatively?

It is unclear who is a participant in the IDI.

The tool is not well addressed in the instrumentation and data collection procedures. (especially the IDI guide).

Please revise your recommendation as per your findings.

6. PLOS authors have the option to publish the peer review history of their article (what does this mean?). If published, this will include your full peer review and any attached files.

Reviewer #1: **Yes: **Tesfaye Yimer Tadesse

Reviewer #2: No

---

## [Author Response · Author response to Decision Letter 0]

10 May 2023

Dear Reviewers and Editors, We are very grateful for your relevant comments, questions and suggestions. Following this, we have provided our responses point by point as presented via response letter.

---

## [Decision Letter · Decision Letter 1]

26 Jun 2023

Assessment of COVID-19 Vaccine Acceptance and its associated factors in Debre Berhan City, Ethiopia, 2022

PONE-D-23-01580R1

Dear Dr. Tefera,

We’re pleased to inform you that your manuscript has been judged scientifically suitable for publication and will be formally accepted for publication once it meets all outstanding technical requirements.

Kind regards,

Ayechew Ademas Tesema, MSc

Academic Editor

PLOS ONE

Additional Editor Comments (optional):

dear the authors although the manuscript is improved and now it needs the following corrections prior to the final decision. The author shall correct some minor correction like font type, color variation across the text, table problem see table 1 and 2, alignment from left side see line 202-203, change Contact

"someone with COVID-19 patient" by "do you have contact with a COVID-19 patient ?"

Reviewers' comments:

Reviewer's Responses to Questions

**Comments to the Author**

1. If the authors have adequately addressed your comments raised in a previous round of review and you feel that this manuscript is now acceptable for publication, you may indicate that here to bypass the “Comments to the Author” section, enter your conflict of interest statement in the “Confidential to Editor” section, and submit your "Accept" recommendation.

Reviewer #1: All comments have been addressed

2. Is the manuscript technically sound, and do the data support the conclusions?

Reviewer #1: Yes

3. Has the statistical analysis been performed appropriately and rigorously? 

Reviewer #1: Yes

4. Have the authors made all data underlying the findings in their manuscript fully available?

Reviewer #1: Yes

5. Is the manuscript presented in an intelligible fashion and written in standard English?

Reviewer #1: Yes

6. Review Comments to the Author

Reviewer #1: The authors addressed the comments and suggestions raised in the first round of the review process. The manuscript is amended, and it meets the criteria of publications and is ready to be published. I suggest the editor to accept the manuscript.

7. PLOS authors have the option to publish the peer review history of their article (what does this mean?). If published, this will include your full peer review and any attached files.

Reviewer #1: No

---

## [Editor Report · Acceptance letter]

7 Jul 2023

PONE-D-23-01580R1 

Assessment of COVID-19 Vaccine Acceptance and its associated factors in Debre Berhan City, Ethiopia, 2022 

Dear Dr. Tefera:

I'm pleased to inform you that your manuscript has been deemed suitable for publication in PLOS ONE. Congratulations! Your manuscript is now with our production department. 

Kind regards, 

on behalf of

Assistant professor Ayechew Ademas Tesema 

Academic Editor

PLOS ONE